# The Intersection of COVID-19 and Metabolic-Associated Fatty Liver Disease: An Overview of the Current Evidence

**DOI:** 10.3390/v15051072

**Published:** 2023-04-27

**Authors:** Mykhailo Buchynskyi, Iryna Kamyshna, Valentyn Oksenych, Nataliia Zavidniuk, Aleksandr Kamyshnyi

**Affiliations:** 1Department of Microbiology, Virology, and Immunology, I. Horbachevsky Ternopil National Medical University, 46001 Ternopil, Ukraine; 2Department of Medical Rehabilitation, I. Horbachevsky Ternopil National Medical University, 46001 Ternopil, Ukraine; kamyshna_ii@tdmu.edu.ua; 3Department of Clinical and Molecular Medicine (IKOM), Norwegian University of Science and Technology (NTNU), 7028 Trondheim, Norway; valentyn.oksenych@ntnu.no; 4Department of Infectious Diseases with Epidemiology, Dermatology and Venerology, I. Horbachevsky Ternopil National Medical University, 46001 Ternopil, Ukraine; zavidnyuk_ng@tdmu.edu.ua

**Keywords:** fatty liver disease, MAFLD, coronavirus disease 2019, SARS-CoV-2

## Abstract

The global population is currently experiencing the impact of the SARS-CoV-2 coronavirus, which has caused the Coronavirus Disease 2019 (COVID-19) pandemic. With our profound comprehension of COVID-19, encompassing the involvement sequence of the respiratory tract, gastrointestinal system, and cardiovascular apparatus, the multiorgan symptoms of this infectious disease have been discerned. Metabolic-associated fatty liver disease (MAFLD), formerly known as non-alcoholic fatty liver disease (NAFLD), is a pervasive public health concern intricately linked with metabolic dysregulation and estimated to afflict one-fourth of the global adult population. The burgeoning focus on the association between COVID-19 and MAFLD is justified by the potential role of the latter as a risk factor for both SARS-CoV-2 infection and the subsequent emergence of severe COVID-19 symptoms. Investigations have suggested that changes in both innate and adaptive immune responses among MAFLD patients may play a role in determining the severity of COVID-19. The remarkable similarities observed in the cytokine pathways implicated in both diseases imply the existence of shared mechanisms governing the chronic inflammatory responses characterizing these conditions. The effect of MAFLD on the severity of COVID-19 illness remains uncertain, as indicated by conflicting results in cohort investigations.

## 1. Introduction

The global COVID-19 pandemic, initiated by the emergence of a novel virus that was subsequently identified as SARS-CoV-2, came to the fore in December 2019, prompting a profound and ongoing impact on global society [1]. As of April 2023, the global count of confirmed COVID-19 cases had exceeded 750 million, with more than 6.8 million fatalities [2]. Mortality rates from severe COVID-19 range from 21% to 30% [3], underscoring the critical importance of exploring the link between this disease and other disorders.

Current evidence points towards an association between severe COVID-19 outcomes and factors such as older age [4,5,6,7], male gender [4,6,7], and multiple comorbidities including hypertension [8,9], cardiovascular disease [7], obesity [4,5,10,11], and type 2 diabetes [7,12,13,14,15].

According to global statistics, the prevalence of metabolic disorders is on the rise, with an estimated 422 million adults worldwide living with diabetes in 2014 [16]. In addition, obesity rates have tripled since 1975, with more than 1.9 billion adults worldwide being overweight or obese in 2016 [17].

Vascular disorders are also a significant health concern globally, with hypertension affecting approximately 1 billion people worldwide [17]. Atherosclerosis, a condition in which fatty deposits accumulate in the arteries, is the underlying cause of most cardiovascular diseases and is prevalent worldwide [18].

The prevalence of metabolic and vascular disorders in those who succumbed to COVID-19 is considerable, accounting for up to 50% of fatalities [19]. The interdependence between COVID-19 and the metabolic and endocrine systems is increasingly evident, pointing towards a bidirectional relationship. Individuals with metabolic disorders such as obesity, hypertension, diabetes, and non-alcoholic fatty liver disease are at an elevated risk of severe COVID-19. Conversely, SARS-CoV-2 infection can trigger the onset of diabetes or exacerbate existing metabolic disorders [15,19].

The risk of severe COVID-19 is increased in individuals with obesity due to several underlying mechanisms, including heightened inflammation, hyper coagulopathy, and mechanical obstruction [20]. Additionally, obesity and diabetes correlate with an elevated risk of pulmonary fibrosis, chronic obstructive pulmonary disease, and reduced respiratory function [21].

The probability of experiencing stroke and cardiovascular complications is increased in those with obesity, diabetes, and hypertension [22]. The overexpression of prothrombotic factors such as coagulation factors (II, VII, VIII, IX, XI, and XII), PAI-1 (plasminogen activator inhibitor-1), and the von Willebrand factor in patients contributes to an augmented coagulation response. In conjunction with pre-existing factors, these risk factors can lead to an increased likelihood of a stroke or pulmonary embolism [23,24].

Metabolic dysfunction is often accompanied by chronic inflammation, which is characterized by an upregulation in the release of proinflammatory cytokines, including TNFα, IL-6, and IL-1β, among affected individuals. In patients with metabolic syndrome, these cytokines are released by immune cells in adipose tissue, leading to the suppression of insulin signaling [25].

As a consequence of increased cytokine activity, the production of leptin and PAI-1 is amplified, whereas adiponectin secretion is reduced, culminating in immune cell and macrophage infiltration in tissues, including white adipose tissue, skeletal muscle, the liver, and the pancreas [25]. Insulin resistance provokes the infiltration of predominantly M1 macrophages in adipose tissues [26].

In contrast to non-obese individuals, obese ones exhibit sustained IL-6 receptor expression, which contributes to the development of a chronic low-grade inflammatory state known as meta-inflammation [27]. The impairment of insulin signaling exacerbates the state of chronic inflammation through the activation of AP-1 and NF-kB, resulting in a reduction of anti-inflammatory cytokines and an elevation of pro-inflammatory cytokines (TNFα, IL-6, and IL-1β). The switch from an anti-inflammatory to a pro-inflammatory status aggravates insulin resistance [28], leading to airway hyperreactivity and augmenting the risk of respiratory failure and cardiopulmonary collapse in those afflicted with diabetes and COVID-19 [26].

Metabolic-associated fatty liver disease (MAFLD) is a hepatic disorder characterized by metabolic dysfunction and the presence of hepatic steatosis, diagnosed by histologic or imaging evidence, in addition to at least one of the following: type 2 diabetes mellitus (T2DM), overweight/obesity, or another metabolic dysfunction. An important feature of the updated MAFLD definition is the removal of the previous requirement to exclude significant alcohol consumption or other chronic liver diseases in order to establish a diagnosis [29,30]. A new definition of MAFLD may better describe the hepatic manifestation of metabolic syndrome than the traditional definition of non-alcoholic fatty liver disease (NAFLD) [31,32,33]. The adoption of the term MAFLD, replacing NAFLD, represents a promising opportunity to mitigate the unintended negative impacts associated with the older term, such as stigmatization, trivialization, and limited patient awareness. Furthermore, this shift in terminology may serve as a catalyst for increasing funding and prompting health policy action [29].

The global prevalence of MAFLD is currently estimated to be around 25%, making it the most frequent cause of chronic liver disease, liver cirrhosis, and even hepatocellular carcinoma [34].

In contrast to the well-established cardiac as well as pulmonary and gastrointestinal manifestations associated with SARS-CoV-2 infection, the clinical implications of liver involvement have remained a topic of debate since the onset of the COVID-19 pandemic [35,36,37,38,39,40,41].

This paper provides a comprehensive overview of the epidemiology of COVID-19 in patients diagnosed with MAFLD. The discussion encompasses the underlying mechanisms and potential explanations for any observed interactions between the two conditions.

The majority of studies utilized the definition of NAFLD rather than MAFLD. We endeavored to maintain fidelity to the outlined descriptions provided in relevant studies by retaining the old NAFLD abbreviation in relevant cases.

## 2. The Role of MAFLD in the Progression of COVID-19

Individuals with MAFLD demonstrate disrupted hepatic immune function [42,43]. Chronic inflammation, a key factor in the pathogenesis of fatty liver disease, is modulated by the actions of hepatic and adipose tissue macrophages through the secretion of cytokines and adipokines [43].

Obesity, commonly linked with MAFLD, can induce the transformation of anti-inflammatory M2 macrophages into pro-inflammatory M1 macrophages through the process of polarization. This unique macrophage transition is mediated by the presence of bile acids, resulting in enhanced lipid accumulation, as well as the development of both local and systemic low-grade chronic inflammation [44].

Circulating levels of IL-6 are significantly elevated in individuals diagnosed with MAFLD [45]. The contribution of the inflammatory response to the severity of COVID-19 has been well-established in previous research [46,47,48] with clinical deterioration in certain infected patients linked to a virus-induced cytokine “storm” [49]. The presence of MAFLD augments the cytokine storm induced by the virus through the release of a multitude of pro-inflammatory cytokines, including IL-6 [45].

The liver is enriched with various innate immune cells, and the presence of fatty infiltration alters its immune response toward increased inflammation [44]. The presence of nonalcoholic steatohepatitis (NASH) is associated with a constant low-grade inflammatory response involving cytokine recruitment, oxidative stress, mitochondrial dysfunction, and endoplasmic reticulum dysfunction. Its presence in patients with COVID-19 may exacerbate the virus-induced cytokine storm by releasing many inflammatory mediators from the liver [50].

Several studies of varying quality have investigated the association between non-alcoholic fatty liver disease and the risk of morbidity and mortality due to COVID-19 (Table 1). The majority of available studies on the association between NAFLD/MAFLD and COVID-19 are retrospective and have limited sample sizes. Moreover, the literature has considerable heterogeneity in the definitions of NAFLD [51] and its updated counterpart, MAFLD. Some studies have relied solely on surrogate markers for hepatic steatosis and fibrosis, such as the hepatic steatosis index, NAFLD fibrosis score, or Dallas steatosis index [52,53,54,55,56,57,58,59,60,61,62,63].

Meanwhile, imaging techniques such as ultrasound or computed tomography (CT) are the primary tools used to identify the presence of fatty liver [57,64,65,66,67,68,69,70,71,72,73,74,75,76,77,78]. The use of a liver biopsy to confirm NAFLD is a rare occurrence in the literature [74]. Furthermore, inconsistent definitions of severe COVID-19 progression have been employed in various studies. It should be noted that the use of blood-based surrogate scores or imaging techniques (ultrasound, CT) during hospitalization for COVID-19 to diagnose NAFLD does not provide information about the presence of fatty liver before the emergence of COVID-19. Control groups often have fewer patients with classic metabolic factors, such as diabetes mellitus and obesity, compared to the corresponding NAFLD groups [57,75,78]. As a result, this metabolic imbalance among study groups cannot be easily addressed through multivariate analysis.

**Table 1 viruses-15-01072-t001:** Research studies examining the relationship between MAFLD and the risks of incidence, severity, and mortality associated with COVID-19.

References	Was Used in Meta-Analysis (Total Number)	Contry of Study	Study Design, Included Patients (Total/NAFLD, MAFLD)	Outcomes	Limitations
Zhou et al., 2020 [64]	[79,80,81,82,83] (5)	China	Retrospective, matched cohorts, 110/55	Independent of other confounding factors, the presence of MAFLD among patients below 60 years of age is positively associated with the development of severe or critical COVID-19.	Matching of patients was not performed based on the primary outcome variable. Asian cohort.
Zhou et al., 2020 [71]	[79,80,81,82,84] (5)	China	Retrospective cohort study, 327/93	Younger patients with MAFLD have a higher risk for severe COVID-19 progression or mortality.	A minor sample size of the older cohort of patients. Asian cohort.
Zheng et al., 2020 [72]	[80,81,83] (3)	China	Retrospective cohort study, 214/66	Co-occurring obesity in patients with MAFLD was found to increase the risk of severe illness by over six times.	Patients did not undergo liver biopsy. Waist circumference was not measured in patients. Patients were only of Asian ethnicity.
Huang et al., 2020 [52]	[79,80,83] (3)	China	Retrospective cohort study, 280/66	SARS-CoV-2 infection in patients NAFLD is positively associated with an elevated risk of liver injury development. However, no patient with COVID-19 with NAFLD developed severe liver injury.	HSI was employed for the purpose of identifying the presence of NAFLD in the absence of any known liver pathologies. Asian cohort.
Ji et al., 2020 [53]	[79,80,81,82,83,84] (6)	China	Retrospective cohort study, 202/76	Injury in patients with COVID-19 was frequent but mild in nature.	Small sample size; the Asian cohort. Very different co-morbidities among groups, definition of NAFLD only through IHS.
Targher et al., 2020 [56]	[79,80,81,82,83] (5)	China	Retrospective cohort study, 310/94	More severe COVID-19 with higher FIB-4 or NFS.	Small sample size; the Asian ancestry of the cohort and the use of NFS without a histological diagnosis of liver fibrosis. No full paper.
Gao et al., 2021 [73]	[79,80,81,82,83] (5)	China	Retrospective case-control study, 130/65	The presence of MAFLD in nondiabetic patients was associated with a four-fold increased risk of severe COVID-19.	Diagnosing NAFLD only by CT and clinical criteria. Same patients with Zhou et al., 2020 [64].
Chen et al., 2021 [57]	[80,83] (2)	USA	Retrospective single-center cohort study, 342/178	The presence of HS in COVID-19 patients was observed to correlate with augmented disease severity and transaminitis.	Comorbidities were not taken into account. Metabolic status is not balanced. Using the HSI and imaging to define HS.
Hashemi et al., 2020 [74]	[79,80,84] (3)	USA	Retrospective cohort study,363/55	NAFLD significantly associated with ICU admission and with needing mechanical ventilation.	Using both imaging studies and histopathology for diagnosing LCD. Patients with milder courses of COVID-19, potentially over-estimating the effects of SARS-CoV-2 on the liver
Bramante et al., 2020 [58]	[79,80,83] (3)	USA	Retrospective cohort study, 6400/373	COVID-19 hospitalization is significantly associated with the presence of NAFLD/NASH, and this risk appears to be attributable to obesity.	The presence of unmeasured confounders and residual bias may impact the validity of the results.
Kim et al., 2021 [59]	[80,82] (2)	USA	Multicenter Observational cohort study, 867/456	NAFLD does not have any risk factors for severe progression or mortality of COVID-19.	No control cohort with liver disease, NAFLD ICD diagnosis.
Steiner et al., 2020 [75]	[84] (1)	USA	Cross-sectional study, 396/213	The likelihood of severe COVID-19 manifestation was higher among patients with NAFLD.	NAFLD was defined through imaging. Lack of information about metabolic status, no full paper.
Marjot et al., 2021 [85]	[82] (1)	UK, USA	Retrospective cohort study, 932/362	Patients with AIH were the same risk-averse outcomes as CLD causes (including NAFLD).	No clear NAFLD definition. Compared only AIH and CLD cohorts.
Marjot et al., 2021 [51]	[83] (1)	UK (origin) but data is multinational	Retrospective cohort study, 1345/322	No increased mortality of patients with NAFLD.	There were no specific cohorts for NAFLD patients. No words about NAFLD definition.
Forlano et al., 2020 [76]	[80,82,83,84] (4)	UK	Retrospective cohort study, 193/61	The presence of NAFLD was not associated with worse outcomes in patients with COVID-19. NAFLD patients were younger on admission.	Study population was small. Only visualization methods were used.
Vázquez-Medina et al., 2022 [60]	[82] (1)	Mexico	Retrospective cohort study, 359/NAFLD–79, MAFLD–220.	The MAFLD cohort displayed a higher fatality rate, whereas the NAFLD group did not exhibit any marked distinction.	Using a noninvasive method for defining NAFLD and MAFLD.
Moctezuma-Velázquez et al., 2021 [77]	[82] (1)	Mexico	Retrospective cohort study, 470/359	NAFLD as per the DSI was associated with death and IMV-need in hospitalized patients with COVID-19.	Definition NAFLD based on DSI, CT.
Lopez-Mendez et al., 2020 [61]	[80,83] (2)	Mexico	Retrospective, cross sectional study,155/66	Prevalence of HS and significant liver fibrosis was high in COVID-19 patients but was not associated with clinical outcomes.	Estimating liver fibrosis through non-invasive models.
Mahamid et al., 2021 [78]	[79,80,82,83,84] (5)	Israel	Retrospective case-control study, 71/22	The risk of severe COVID-19 was elevated in patients with NAFLD, regardless of gender and irrespective of the presence of metabolic syndrome.	Differences in metabolic status between groups, the small number of COVID-19 patients underwent CT to diagnose NAFLD
Mushtaq et al., 2021 [62]	[84] (1)	Qatar	Retrospective cohort study, 269/320	The presence of NAFLD is a predictor of mild or moderate liver injury, but not for mortality or COVID-19 severity.	Using HSI index for diagnosing. No full paper.
Parlak et al., 2021 [65]	[83] (1)	Turkey	Retrospective cohort study, 343/55	NAFLD was an independent risk factor for COVID-19 severity.	Definition NAFLD based on CT. Missed data comparing NAFLD and non-NAFLD cohorts
Yoo et al., 2021 [63]	[82] (1)	South Korea	Retrospective cohort study, 72,244/54,913 (HIS—26,041, FLI—19,945, claims-based—8927).	Patients with pre-existing NAFLD had a higher likelihood of severe COVID-19 illness.	Using HIS, FLI, and claims-based NAFLD for defining fat liver for the same patients.
Vrsaljko et al., 2022 [66]	[82] (1)	Croatia	Prospective cohort study, 216/120	NAFLD is associated with higher COVID-19 severity, more adverse outcomes, and more frequent pulmonary thrombosis.	Abdominal ultrasound was employed for the diagnosis of NAFLD.
Valenti et al., 2020 [86]	N/A	UK	Mendelian randomization, 1460/526	The predisposition for severe COVID-19 is not directly augmented by a genetic propensity for hepatic fat accumulation.	These results were obtained in an initial set of cases without detailed characterization. No full paper.
Liu et al., 2022 [87]	N/A	UK	Mendelian randomization, N = 2,586,691	No evidence to support a causal relationship between COVID-19 susceptibility/severity and NAFLD.	Potential data errors, limited patient characterization. Missing information about NAFLD cohort. No full paper
Roca-Fernández et al., 2021 [88]	N/A	UK	Prospective cohort study (UK Biobank), 1043/327	Patients with fatty liver disease were at increased risk of infection and hospitalization for COVID-19.	Small proportion of UKB participants. Restriction for blood biomarkers of liver disease.
Simon et al., 2021 [67]	N/A	Sweden	Matched cohort study using the ESPRESSO, 182,147/42,320-LCD, 6350-NAFLD	Patients with CLD had a higher risk of hospitalization for COVID-19 but did not have an increased risk of severe COVID-19.	There is no comparison for NAFLD cohort. Not every CLD was confirmed though biopsy. The cohort lacked detailed data regarding body mass index or smoking.
Chang et al., 2022 [54]	N/A	South Korea	Retrospective cohort study, 3112–FLI score	An augmented risk of severe COVID-19 complications was observed in patients with high fatty liver index (FLI), reflective of NAFLD.	Using FLI score for determining NAFLD. Dataset did not directly confirm NAFLD through biopsy or ultrasound. The time gap between body measurements in health screening and COVID-19 infection.
Okuhama et al., 2022 [68]	N/A	Japan	Retrospective cohort study, 222/89–fatty liver	The manifestation of fatty liver on plain CT scan at the time of admission may constitute a risk factor for severe COVID-19.	No determination NAFLD/MAFLD. Using CT scan for screening fat liver disease.
Tripon et al., 2022 [55]	N/A	French	Retrospective cohort study, 719/311	Patients with NAFLD disease and liver fibrosis are at higher risk of progressing to severe COVID-19.	Using NFS for determining NAFLD. Missing some important parameters.
Campos-Murguía et al., 2021 [69]	N/A	Mexico	Retrospective cohort study, 432/176	In contrast to the presence of MAFLD, the occurrence of fibrosis is correlated with a heightened risk of severe COVID-19 and mortality.	Liver steatosis was diagnosed by CT scan, and fibrosis by non-invasive scores.
Ziaee et al., 2021 [70]	N/A	Iran	Retrospective cohort study, 575/218	Fatty liver is significantly more prevalent among COVID-19 against non-COVID-19 patients, they develop a more severe disease and tend to be hospitalized for more extended periods.	There was no access to each patient’s past medical history, so the term “fatty liver patients” was used. The lack of diagnosis data for control group patients

HS—hepatic steatosis; MAFLD—metabolic-associated fatty liver disease; NAFLD—non-alcoholic fatty liver disease; CLD—chronic liver disease; HSI—hepatic steatosis index; CT—computer tomography; NFS—non-alcoholic fatty liver disease fibrosis score; ICD—international classification of diseases; AIH—autoimmune hepatitis; FLI—fatty liver index; DSI—Dallas steatosis index; ESPRESSO—epidemiology strengthened by histopathology reports in Sweden.

While the majority of extant studies conclude that MAFLD/NAFLD is associated with an augmented susceptibility to contracting COVID-19, as well as an increased probability of requiring admission to intensive care, its influence on the development of critical COVID-19 or mortality remains unclear.

At present, the available data indicate that the presence of NAFLD alone may not be a significant risk factor for severe COVID-19 progression or mortality. Notably, studies of registries comprised of large liver collectives with diverse etiologies tend to suggest that NAFLD may not have a distinct role in this regard.

Our review also encompasses meta-analyses that investigate the interplay between COVID-19 and MAFLD/NAFLD (Table 2).

**Table 2 viruses-15-01072-t002:** Characteristics of meta-analysis that have investigated the interaction between MAFLD and COVID-19.

Refferences	Number of Studies/Included Patients	Results	Advantages	Limitations
Hegyi et al., 2021 [79]	9 studies (8202 cases)	A 2.6-fold increased risk of severe COVID-19 is associated with MAFLD, while NAFLD is linked to a five-fold greater susceptibility; however, there was no discernable difference in hospital mortality between COVID-19 patients with MAFLD or NAFLD.	The study was executed with meticulous attention to methodological rigor.	Study involves only nine articles. Most of the articles were published in Asian countries. Data came mostly from retrospective studies; in-hospital mortality was not analyzed; and high risk of bias in included articles.
Singh et al., 2021 [80]	14 studies (1851 cases)	In patients with COVID-19 infection, the presence of NAFLD increased the risk of severe disease and ICU admission; however, there was no discernable difference in mortality between COVID-19 patients with or without NAFLD.	The study’s findings were adjusted for several possible confounding factors to provide a more accurate assessment of the relationship between the variables of interest.	The study involved only six articles; There was a lack of a robust and consistent definition of NAFLD in selected articles. Major covariates, such as age, sex, race, and co-morbidities were adjusted in selected studies. There was a restricted scope for a robust subgroup analysis due to a fewer number of included studies.
Tao et al., 2021 [84]	7 studies (2141 cases)	The presence of MAFLD was linked to an elevated risk of severe COVID-19 (odds ratios: 1.80, 95% CI: 1.53–2.13, *p* < 0.00001), but not to an increased likelihood of death due to COVID-19 infection.	The study’s robustness was further substantiated by a sensitivity analysis, which validated the initial findings. The inclusion of studies from both Chinese and foreign countries bolstered the generalizability of the results and improved the external validity of the study.	Insufficient representation of studies, notably those from China, limited the robustness of the meta-analysis. The majority of studies included in this analysis were cross-sectional, which may compromise their reliability compared to more robust cohort studies. The etiology of the variation in the pooled prevalence could not be determined.
Pan et al., 2020 [81]	6 studies (1293 cases)	The study demonstrated that MAFLD is independently associated with an elevated risk of severe COVID-19 and a higher prevalence of COVID-19 in individuals with MAFLD compared to the general population.	The heterogeneity observed in the studies was reasonably acceptable, thereby ensuring the reliability of the outcomes.	The included studies were limited in number and conducted exclusively within China. There were only six studies included in the final analysis; only one study had a subgroup analysis. The studies included in the analysis were mainly cross-sectional and case-control studies, which are generally regarded as less robust than prospective cohort studies.
Wang et al., 2022 [82]	18 studies (22,056 cases)	The presence of NAFLD was found to be independently associated with severe COVID-19, particularly in younger patients compared to older ones.	The overall odds ratio was derived by considering the effects sizes adjusted for risk factors; mainly age, sex, smoking, obesity, diabetes, and hypertension. A sensitivity analysis was conducted (it showed no significant impact on the overall results).	There is no statement regarding protocol and registration. Authors used free-text only in their search strategy without including the MeSH approach. Detailed flow diagram that would illustrate the study selection process, sample size, PICOS, follow-up period, and citations of the included studies was not provided. The authors did not report the odds ratio (OR) in a clear and specific manner. There is no corresponding analysis of the risk of bias. No full paper.
Li et al., 2022 [89]	3 genome-wide association study (8267 cases)	The available evidence does not suggest a direct cause-and-effect association between NAFLD and the severity of COVID-19. The correlation between NAFLD and COVID-19 reported in prior studies is likely explained by the interrelatedness of NAFLD and obesity. The impact of comorbid factors associated with NAFLD on severe COVID-19 is largely attributed to body mass index, waist circumference, and hip circumference, based on evidence of causality.	Mendelian Randomization analysis provides a possibility to examine the causal relationship between NAFLD and severe COVID-19; Using COVID-19 genome-wide association study summary statistics.	The findings of the study cannot be generalized to evaluate the relationship between the severity of NAFLD and the risk of severe COVID-19, as the analysis only considered the presence of liver fat as the exposure variable. One of the selected research studies utilized one-sample-based MR analysis, which may be susceptible to bias [86]. The present findings may be subject to limitations arising from the small size of the sample population, as well as potential confounding clinical covariates that remain unidentified. These results somewhat contradict the observational studies of the same authors in other studies [7,90];
Hayat et al., 2022 [83]	16 studies (11,484 cases)	The occurrence of COVID-19 was found to be 0.29 among individuals with MAFLD. A heightened likelihood of COVID-19 severity and higher ICU admission rate were observed among patients with MAFLD. The correlation between MAFLD and COVID-19 mortality did not achieve statistical significance.	This study represents a novel contribution to the field, as it is the first to comprehensively investigate COVID-19-related mortality in a large and diverse cohort of MAFLD patients. Additionally, the study uniquely examines both the prevalence of MAFLD and the associated COVID-19 outcomes in a broad and extensive MAFLD population.	The respective studies included in this meta-analysis did not include a robust and consistent definition of the severity of COVID-19. Some included studies do not account for confounding factors such as age, race, gender, and certain other co-morbidities. There were multiple comorbidities in the study population, making it difficult to dissect the contribution of each comorbidity to COVID-19 outcomes. Fewer studies were included in the subgroup analysis of the effect of MAFLD on the COVID-19 ICU entrance and mortality rate making it difficult to analyze the publication bias (fewer than ten articles).

In numerous studies, patients with NAFLD were found to have a four-fold higher risk of developing severe COVID-19 compared to the control group [73,91,92]. The meta-analysis by Hegyi et al., 2021—one of the first to address this issue—assessed whether NAFLD is associated with a more severe course of COVID-19, intensive care unit admission, and mortality [79]. The results of the meta-analysis confirm that NAFLD increases the likelihood of developing severe COVID-19 by 2.6 times compared to the control group. Furthermore, analyzing groups with and without NAFLD revealed a five-fold increase in the risk of developing severe COVID-19.

Another independent meta-analysis by Singh et al., 2021, provides similar results [80]. This systematic review aimed to evaluate the clinical outcomes of patients with confirmed COVID-19 and existing NAFLD. Patients with these comorbidities had an increased risk of intensive care unit hospitalization; however, no difference in mortality was observed between patients with COVID-19 with or without underlying NAFLD.

The results of other systematic reviews [81,82,84] highlight similar findings to those of the previous reviews [79,80]. The meta-analysis by Wang et al., 2022 [82] showed that the presence of NAFLD was significantly independently associated with a more severe course of COVID-19 among younger patients aged <60 years but not among older individuals >60 years.

The position of the European Association for the Study of the Liver (EASL) regarding the issue of comorbidity between NAFLD and COVID-19 is indicative: patients with NAFLD have an increased overall risk of developing severe COVID-19, which may be associated with the presence of other high-risk comorbidities [93]. This is consistent with the results of previous meta-analyses [79,80,81,82,84].

Other studies exist that argue for the opposite perspective. Li et al., 2022, conducted a large-scale, two-sample Mendelian randomization analysis (TSMR) [89]. Mendelian randomization uses genetic variations as a natural experiment to investigate causal relationships between potentially modifiable risk factors and health outcomes in observational data. A genome-wide meta-analysis was also conducted to identify single nucleotide polymorphisms associated with NAFLD and investigate the impact of 20 major associated factors with NAFLD on severe COVID-19.

This study examined the causal relationships between NAFLD, serum alanine aminotransferase, degree of steatosis, NAFLD activity score or fibrosis stage, and severe COVID-19. The results of this study did not find any evidence that NAFLD is a risk factor for severe COVID-19 and suggested that the link between NAFLD and COVID-19 is explained by the presence of obesity in this patient cohort.

The study only included works that investigated a cohort of patients of European descent. In the analysis of the results of multiple logistic regression, which evaluated the relationship between eight risk factors (age, male gender, T2D, NAFLD, CVD, liver cirrhosis, and systolic BP) and severe COVID-19, NAFLD was not associated with severe COVID-19 (OR, 1.57; *p* = 0.09). However, this result may be limited by the small sample size, as well as other unknown clinical variables.

Additionally, when investigating the causal relationships between multiple risk factors and COVID-19, obesity indices (BMI, waist circumference, and hip circumference) were the only causally associated risk factors for severe COVID-19, whereas T2D, CVD, SAT, and NAFLD were not. These results contradict the observational studies by the same authors [7,90]. However, they explain this result by the low proportion of variance in severe COVID-19 explained by these factors in this population.

This study shows that not only NAFLD but also T2D, cardiovascular diseases, and other risk factors are not the causes of severe COVID-19. The only factors that were found to be associated with severe COVID-19 were BMI, waist, and hip circumference. These conclusions differ somewhat from previous studies [79,80,81,82,84] and require further discussion in detail.

In summarizing the results of conducted meta-analyses, it can be assumed that the presence of MAFLD/NAFLD increases the risk of severe COVID-19 progression and raises the chances of patients being admitted to intensive care units. However, it does not affect mortality [79,80,81,82,83,84]. The absence of genetic causal connections between the presence of NAFLD and its impact on severe COVID-19 outcomes does not allow for a direct link to be established [89]. This may occur indirectly through the presence of accompanying factors that are components of the metabolic syndrome. We consider that the available data and limitations of previous studies should be taken into account as this topic requires further investigation.

## 3. The Hepatic Implications of COVID-19

The current evidence suggests that aberrations in liver enzymes are frequently observed in individuals with COVID-19 [94]. Liver injury associated with COVID-19 is defined as any damage that occurs to the liver during the COVID-19 disease and its management, regardless of a prior history of liver disease. This injury can be mediated through multiple potential pathomechanisms, such as direct cytotoxicity resulting from active viral replication of SARS-CoV-2 in the liver [95,96], immune-mediated liver injury due to systemic inflammatory response syndrome (SIRS) induced by COVID-19 [97], hypoxic changes due to respiratory failure, vascular alterations associated with coagulopathy [98], endotheliitis or cardiac stasis due to right heart failure [99], drug-induced liver injury [100,101], and exacerbation of underlying liver diseases.

Despite the lack of certainty regarding the precise consequences of COVID-19 on hepatic physiology, it is worth noting that aberrations in liver biochemistry are commonly observed in individuals with COVID-19. In the early stages of the disease, liver biochemistry abnormalities are primarily characterized by mild to moderate elevations of alanine aminotransferase (ALT) and/or aspartate aminotransferase (AST) [102,103,104], intermittent increases in serum bilirubin levels [102,105,106,107,108,109,110], and decreased serum albumin levels [105,106,107,109,111,112,113,114,115], with an infrequent elevation of markers of bile duct damage such as alkaline phosphatase (ALP), gamma-glutamyltransferase (GGT), and total bilirubin (TBIL) (Table 3).

**Table 3 viruses-15-01072-t003:** Biochemical liver abnormalities in patients affected by COVID-19.

Study	Region	Sample Size (n)	Elevated ALT	Elevated AST	Elevated ALP	Elevated GGT	Elevated TBIL	Elevated LDH	Reduced Albumin
Guan et al., 2020 [102]	Nationwide, China	757	158/741 * (21.3%)	168/757 * (22.2%)	N/A	N/A	76/722 * (10.5%)	277/675 * (41.0%)	N/A
Xu et al., 2020 [105]	Hubei, Zhejiang, Anhui, Shandong, Jiangsu, China	581	95/504 * (19%)	82/460 * (18%)	N/A	N/A	60/430 * (14%)	171/383 * (45%)	260/581 * (45%)
Yang et al., 2020 [106]	Hubei Province, China	200	44 (22%)	74 (37%)	N/A	N/A	N/A	74/189 * (38.5%)	144 (72%)
Cai et al., 2020 [116]	Shenzhen, China	417	54 (12.9%)	76 (18.2%)	101 (24.2%)	68 (16.3%)	99 (23.7%)	N/A	N/A
Richardson et al., 2020 [117]	New York, America	5700	2176 (39.0%)	3263 (58.4%)	N/A	N/A	N/A	N/A	N/A
Wang et al., 2020 [107]	Fujian Province, China	199	22 (11.1%)	47 (23.6%)	N/A	N/A	34 (17.%)	65 (32.7%)	26 (13.1%)
Hu et al., 2020 [108]	Hunan Province, China	213	33 (15.5%)	27 (12.7%)	N/A	N/A	44 (20.7%)	27 (12.7%)	N/A
Xiong et al., 2020 [118]	Wuhan, China	116	23 (19.8)	46 (39.7)	N/A	N/A	N/A	69 (59.5%)	N/A
Yang et al., 2020 [109]	Wenzhou, China	149	18 (12.08%)	27 (18.12%)	N/A	N/A	4 (2.68%)	45 (30.20%)	9 (6.04%)
Yu et al., 2020 [111]	Wuhan, China	1445	298/1445 * (20.6%)	303/1445 * (21.0%)	N/A	N/A	N/A	1110/1444 * (76.9%)	723/1443 * (50.1%)
Shen et al., 2020 [110]	Shanghai, China	325	53 (16.3%)	54 (16.6%)	N/A	N/A	N/A	125 (38.5%)	N/A
Zhang et al., 2021 [5]	Wuhan, China	267	49 (18.4%)	76 (28.5%)	N/A	N/A	N/A	N/A	N/A
Xu et al., 2021 [112]	Shanghai, China	1003	295 (29.4%)	176 (17.5%)	26 (2.6%)	134 (13.4%)	40 (4.0%)	N/A	307 (30.6%)
Ding et al., 2021 [119]	Wuhan, China	2073	501 (24.2%)	545 (26.3%)	165 (8.0%)	443 (21.4%)	71 (3.4%)	N/A	N/A
Fu et al., 2021 [113]	Wuhan, China	482	96 (19.9%)	98 (20.3%)	N/A	N/A	23 (4.8%)	N/A	199 (41.3%)
Lv et al., 2021 [114]	Wuhan, China	2912	662 (22.7%)	221 (7.5%)	135 (4.6%)	536 (18.4%)	52 (1.8%)	N/A	2086 (71.6%)
Benedé-Ubieto et al., 2021 [120]	Madrid, Spain	799	204 (25.73%)	446 (49.17%)	186 (24.21%)	270 (34.62%)	N/A	400 (55.84%)	N/A
Weber et al., 2021 [115]	Munich, Germany	217	59 (27.2%)	91 (41.9%)	22 (10.1%)	80 (36.9%)	10 (4.6%)	N/A	71 (32.7%)
Liu et al., 2021 [121]	Changsha, China.	209	20 (9.6%)	24 (11.5%)	N/A	N/A	179 (85.6%)	30 (14.4%)	N/A
Lu et al., 2022 [122]	Sichuan, China	70	32 (45.7%)	22 (31.4%)	12 (17.1%)	32 (45.7%)	32 (45.7%)	40/69 ***** (58.0%)	N/A
Krishnan et al., 2022 [123]	Baltimore, MD, United States	3830	2698 (70.4%)	1637 (44.4%)	611 (16.1%)	N/A	221 (5.9%)	N/A	N/A

* The number of patients with increased liver enzyme levels and decreased albumin/sample size.

The preceding data indicates that the primary target of hepatic injury is hepatocytes. In severe COVID-19 cases, both AST and ALT show significant elevations, accompanied by a mild increase in bilirubin levels [95]. A recent meta-analysis reported an overall prevalence of 20–22.5% and 14.6–20.1% for AST and ALT, respectively [124,125], beyond the reference range; slightly elevated total bilirubin levels were observed in 35% of cases [124]. Although initially believed to be rare [126], subsequent systematic reviews demonstrated that elevated levels of cholestatic liver enzymes, including ALP and GGT, were present in 6.1% and 21.1% of COVID-19 patients, respectively [124,125].

In the study by Bernal-Monterde et al., 2020, which investigated the relationship between COVID-19 and liver injury, an initial increase in transaminases followed by cholestasis was reported. This result may reflect cholestasis at the hepatocellular/canalicular level induced by systemic inflammation, or more severe involvement of the bile ducts in the late stage of the disease [127].

Factors contributing to liver injury and elevated levels of liver enzymes in COVID-19 include immune-mediated inflammatory response, drug-induced liver injury, hepatic congestion, the extrahepatic release of transaminases [128], or direct hepatocyte injury [129].

It has been shown that patients with liver cirrhosis and to a lesser extent, transplanted livers, who were infected with SARS-CoV-2 had an increased risk of mortality [74,130]. Alfishawy et al., 2020 [130] conducted a study that revealed a higher mortality rate (20%) in organ transplant recipients afflicted with SARS-CoV-2 infection, compared to the reported mortality rate of 4–14% among the general population with COVID-19. The authors posit that an increased mortality risk in patients with transplants is associated with advancing age, a higher burden of comorbidities, and the utilization of immunosuppressive therapy.

This is also supported by the meta-analysis by Wang et al., 2022 [131], the results of which indicated that liver cirrhosis is an independent predictor of mortality from COVID-19. These findings are also consistent with the EASL (European Association for the Study of the Liver) document on the latest updates in the treatment of chronic liver diseases [93].

It should also be noted that a liver biopsy study in a group of 48 deceased patients with COVID-19 revealed a large thrombosis of the lumen of vessels at the portal and sinusoidal levels, accompanied by significant pericyte activation and portal fibrosis [98]. Another liver biopsy study of deceased patients with COVID-19 showed moderate microvascular steatosis and moderate lobular and portal inflammatory activity, indicating that the damage may have been a consequence of SARS-CoV-2 infection [132].

An in situ hybridization analysis detected SARS-CoV-2 virions in samples from the lumen of vessels and endothelial cells of the portal vein in patients with COVID-19 [98]. In addition, electron microscopy analyses of liver samples from two deceased patients with elevated liver enzymes after COVID-19 infection revealed intact viral particles in the cytoplasm of hepatocytes [95].

Several drugs are clinically employed to combat COVID-19, including antiviral agents such as remdesivir [133], lopinavir/ritonavir, and interferons [134,135]; antibiotics such as macrolides; antimalarial/antirheumatic drugs such as hydroxychloroquine; immunomodulatory drugs like corticosteroids and tocilizumab; and anti-fever medications such as acetaminophen [136]. However, many of these drugs have been associated with hepatotoxicity [137,138]. The use of lopinavir and ritonavir is independently linked with elevated levels of ALT/AST in COVID-19 patients [139]. The co-occurrence of underlying metabolic abnormalities and MAFLD can contribute to drug-induced liver injury (DILI) [53]. Conversely, MAFLD may also exacerbate the hepatotoxicity of drugs like acetaminophen, leading to the progression of MAFLD to non-alcoholic steatohepatitis (NASH) and even cirrhosis [137]. Corticosteroids, which are recommended for the treatment of severe COVID-19, have also been associated with steatosis [138]. The use of drugs with high hepatotoxicity may therefore contribute to the progression of MAFLD.

The liver is the main organ of metabolism and detoxification in the human body, and even moderate loss of its function can reduce the therapeutic efficacy of antiviral drugs that are metabolized in it. It is important to note that different antiviral drugs may have different pathways of metabolism and elimination, and the degree to which liver dysfunction affects drug metabolism may vary depending on the specific drug. However, liver dysfunction can modify the pharmacokinetics (absorption, distribution, metabolism, and elimination) of many drugs, including antiviral drugs, leading to potential changes in their therapeutic efficacy. Therefore, it is extremely important to better understand the causes of liver damage associated with COVID-19.

## 4. Liver Susceptibility to SARS-CoV-2 Infection

The members of the Coronaviridae family, including SARS-CoV-2, SARS-CoV, and MERS-CoV, are enveloped viruses with a single-stranded RNA of approximately 30 Kb in size, and the angiotensin-converting enzyme 2 (ACE2) receptor is the primary attachment site for SARS-CoV-2 on the cell surface [138].

After attachment, the viral S protein interacts with transmembrane serine protease 2 (TMPRSS2) and preferentially enters the cell by endocytosis, and the viral genome is released from the endosome. Receptor-mediated endocytosis is one of the main mechanisms by which coronaviruses enter the host cell; however, there may be other mechanisms by which the virus can enter the cell depending on the specific virus and host cell types involved. From the two viral polyproteins (pp1a and pp1ab), 16 non-structural proteins (from nsp1 to nsp16) are formed, which serve as building blocks for the virus replication-transcription complex (RTC). The full viral genome replicates in vesicles containing the RTC. Simultaneously, in the Golgi complex, a set of specific subgenomic mRNA is generated for the production of the nucleocapsid and viral envelope of SARS-CoV, which will ensure the subsequent release of mature virions [138].

The widespread expression of the primary entry receptor for the virus, ACE2, may explain how SARS-CoV-2 causes damage to many organs and systems including the intestine, heart, kidneys, pancreas, liver, muscular, and nervous systems [140].

In a healthy liver, the biliary epithelium appears to have the highest expression of ACE2 receptors. Studies conducted using liver-derived and induced pluripotent stem cell (iPSC)-derived organoids suggest that cholangiocytes are highly susceptible to both entry and replication of SARS-CoV-2 [141,142]. Despite this, the observed pattern of liver abnormalities associated with SARS-CoV-2 infection does not align with cholestatic liver injury (Table 3). Hepatocytes, in contrast, express low levels of ACE2, which suggests a potentially lower risk of SARS-CoV-2 entry. However, in vivo, electron microscopy findings indicate the presence of intracellular virus particles within the hepatocyte, accompanied by mitochondrial swelling and structural damage. This strongly suggests direct cytopathy of SARS-CoV-2 in hepatocytes [95,143].

In the context of chronic liver disease and NAFLD, there is a significant increase in the expression of ACE2 receptors [144,145,146]. However, other studies have investigated the impact of MAFLD on the expression of ACE2 receptors and TMPRSS2 in the liver and found no association between MAFLD and changes in the expression of these genes [147,148].

The use of ACE inhibitors stimulates an increase in the expression of ACE2 receptors. Treatment of liver disease and metabolic syndrome with ACE inhibitors may promote increased susceptibility to SARS-CoV-2 and increased severity of COVID-19. However, the results of the study by Cai et al., 2020 [116] showed that in hypertensive patients receiving ACE inhibitors/angiotensin receptor blockers (ARBs), there was no increase in the frequency of COVID-19 progression to a severe form compared to patients taking other antihypertensive drugs. It could be postulated that the administration of ACE inhibitors may result in a negligible expression of ACE2 receptors, or that such expression may not exert a substantial impact on the ability of SARS-CoV-2 to penetrate host cells.

## 5. Imbalance of Intestinal Microbiota

The composition of the intestinal microbiota (IM) is characterized by the presence of numerous species belonging to four predominant bacterial phyla, namely Firmicutes, Actinobacteria, Bacteroidetes, and Gammaproteobacteria. The IM is known to play a pivotal role in the development of NAFLD by exerting a negative effect on tight junction protein expression. As a consequence, increased intestinal permeability occurs, allowing for the translocation of bacterial endotoxins from the intestinal lumen into the systemic circulation [149]. The presence of endotoxins creates an inflammatory milieu by inducing the expression of pro-inflammatory cytokines, hepatic toll-like receptor 4 (TLR4), and plasma plasminogen activator inhibitor 1. This inflammatory response contributes to the development of insulin resistance (IR) and hepatic lipid accumulation. Furthermore, the fermentation of non-digestible carbohydrates by intestinal microbiota leads to the production of bioavailable substrates that enhance the synthesis of fatty acids (FA) and mitigate fasting-induced adipocyte factors within intestinal cells. This process inhibits the activity of lipoprotein lipase, which drives the accumulation of triglycerides (TG) in adipose tissue [149].

The gut microbiota and its metabolites, particularly those possessing immunomodulatory properties, are capable of exerting an influence on the manifestations of COVID-19. Specifically, dysbiosis of the gut microbiota within this context could exacerbate inflammation and various symptoms via its capacity to modulate ACE2 expression in enterocytes and alter the secretion of immunomodulatory compounds, including tryptophan, short-chain fatty acids (SCFAs), and secondary bile acids. Such dysbiosis may also contribute to the development of cytokine storms, which may result in the manifestation of more severe symptoms. Furthermore, in the long term, dysbiosis may be associated with the persistence of COVID-19 symptoms and inflammation, which is referred to as post-acute COVID-19 syndrome (PACS) [150].

The intestinal microbiota participates in diverse metabolic transformations of bile acids, which, in turn, modulate the immune response and promote either pro- or anti-inflammatory effects [151]. Likewise, research has demonstrated the ability of secondary bile acids to suppress NF-κB signaling pathways, impede the development of IL-17-expressing helper T cells, and facilitate the differentiation of regulatory T cells [152]. Concerning COVID-19, a significant correlation was observed between secondary bile acids, the progression of respiratory failure, and patient survival [153].

Patients with COVID-19 exhibited a decrease in anti-inflammatory bacteria, such as Eubacterium ventriosum, Faecalibacterium prausnitzii, Roseburia, and Lachnospiraceae, while opportunistic pathogens, including Clostridium hathewayi, Actinomyces viscosus, and Bacteroides nordii, demonstrated an increase [154]. Similarly, an increased abundance of opportunistic pathogens, including Streptococcus, Rothia, Veillonella, and Actinomyces, and a decreased abundance of beneficial symbionts were observed in patients afflicted with COVID-19 [155].

An additional study, featuring a larger cohort, revealed that several gut commensals, including Faecalibacterium prausnitzii, Eubacterium rectale, and several bifidobacterial species, that possess established immunomodulatory potential, were depleted in COVID-19 patients [156].

Numerous investigations have documented the noteworthy impact of alterations in gut microbiota and their associated metabolites, such as lipopolysaccharides (LPS), indole-3-acetic acid (IAA), peptidoglycan, short-chain fatty acids (SCFA), bile acid metabolites, endotoxins, and several others, on the advancement of NAFLD [151,157,158,159,160,161,162,163,164,165,166].

Our findings revealed that there were common dysregulated bacterial species between the two diseases examined, namely Bacteroides, Eubacterium, Faecalibacterium, Coprococcus, Streptococcus, Enterobacteriaceae, Lactobacillus, and Bifidobacterium. This dysregulated intestinal microbiota may increase inflammatory factors through its metabolites (Table 4).

Intestinal dysbiosis can elicit inflammation in both the intestine and liver, which can be attributed to the translocation of endotoxins and bacteria as a result of a rise in intestinal permeability. This occurrence results in an escalated risk of both local and systemic low-grade inflammation and a reduced anti-inflammatory capacity within the intestine; thus, amplifying the severity of COVID-19 and further exacerbating the progression of MAFLD [170,175,176,177].

Our hypothesis posits that COVID-19 pathogenesis and progression are intricately intertwined with the gastrointestinal system and the liver, with particular emphasis on the potential interactions between gut microbiota, MAFLD, and COVID-19. More extensive investigations into COVID-19 patients are necessary to elucidate the underlying pathophysiological mechanisms and identify the most effective interventions for managing gut microbiota, NAFLD, and COVID-19.

## 6. Bile Acid Receptors FXR and TGF5 as Linking Factors in the Immunopathogenesis of COVID-19 and MAFLD

The Farnesoid-X-receptors (FXR) and the G protein bile acid-activated receptor (GPBAR)-1, also referred to as Takeda G-protein-coupled receptor 5 (TGR5), are the two most extensively characterized receptors belonging to the BAR (bile acids-activated receptors) family [178]. Additionally, cells of the innate immune system, including monocytes/macrophages, dendritic cells (DCs), natural killer (NK), and NKT cells, exhibit high expression levels of both receptors [179,180,181].

The activation of both receptors occurs at relatively low concentrations of bile acids [151]. Cholic acid (CA) and chenodeoxycholic acid (CDCA) are synthesized in the human liver from cholesterol breakdown. After conjugation with glycine or taurine, they are transported through the bile duct into the intestine, where they undergo modifications by the intestinal microbiota to generate secondary bile acids, including deoxycholic acid (DCA) and lithocholic acid (LCA). These modifications are part of a series of complex reactions, and the resulting bile acid profile can differ between individuals based on their diet and gut microbiota composition [182,183,184].

The activation of BARs in macrophages, dendritic cells, and natural killer T cells leads to numerous regulatory functions, which together induce an immunologically tolerant response in the intestine and liver (Figure 1). This response is vital for maintaining tolerance to the constant inflow of dietary antigens and xenobiotics produced by the intestinal microbiota.

Negative regulation of the NF-kB pathway by the Farnesoid-X-receptor (FXR) is achieved through SHP-dependent and -independent mechanisms, leading to counter-regulatory activity on monocytes/macrophages, DCs, and NKT cells [151]. The activation of FXR is reported to decrease the differentiation and activation of intestinal DCs by down-regulating TNF-α expression, which reduces the severity of colitis in two separate mouse model studies. The activation of FXR has been demonstrated to hinder the differentiation of CD14+ monocytes into mature DCs [185,186].

TGR5, similar to FXR, has counter-regulatory effects on the immune response. Activation of TGR5 leads to the transition of colonic macrophages from a pro-inflammatory M1 phenotype to an anti-inflammatory M2 phenotype [182]. The expression of IFN-γ, IL-1β, IL-6, and TNF-α is suppressed by TGR5, whereas IL-10 expression is induced [187].

While NKT cells express both FXR and TGR5, the investigation of activating TGR5 in these cells has been limited to the liver. The activation of TGR5 has been demonstrated to mitigate inflammation by counteracting the polarization of NKT cells towards NKT1, a pro-inflammatory subgroup, and biasing towards NKT10, a regulatory subset of NKT cells that secretes the anti-inflammatory cytokine IL-10 [188].

Stutz et al., 2022 [153], have concluded in their study that elevated concentrations of fecal secondary bile acids are associated with improved outcomes in patients with COVID-19. The authors arrived at a conclusion that the reduction of potential immunomodulatory metabolites, such as secondary bile acids, is positively correlated with the advancement of respiratory failure in individuals afflicted with COVID-19. This result is explained by the immunosuppressive activity of CD4+ regulatory T-cells (Treg). Their numbers are increased by the influence of deconjugated bile acids on them. Additionally, their action on dendritic cells (DCs) has been found to decrease their immunostimulatory properties [189].

The findings suggest that in adult NAFLD patients, dysregulated bile acid (BA) metabolism is associated with an increased risk of hepatic injury [79]. A different investigation has reported that gut microbiota (GM)-mediated deconjugation of bile acids (BAs) stimulates the activation of the farnesoid X receptor (FXR) signaling pathway in the intestine, leading to reduced expression of the cholesterol 7 alpha-hydroxylase (CYP7A1) enzyme and inhibition of the FXR-small heterodimer partner (SHP) pathway. These events culminate in the acceleration of lipid synthesis and the subsequent development of liver disease [190].

All of these data suggest that dysregulated immune responses resulting from altered regulation by bile acids due to changes in gut microbiota composition lead to increased inflammation in the pathogenesis of both COVID-19 and MAFLD. In the pathogenesis of fatty liver disease, bile acids play an even more significant role due to their additional impact on lipid and glucose metabolism [191].

## 7. Conclusions and Perspectives

Although both MAFLD and COVID-19 have spread in a pandemic manner, their progression differs substantially. While infectious diseases typically induce short-term illnesses, MAFLD represents a chronic pandemic. Notably, the molecular mechanisms of inflammation are similar, with MAFLD being characterized by persistent low-grade inflammation and COVID-19 by an acute inflammatory state. Recent studies have enabled a greater understanding of the interaction between these conditions and the potential consequences of their comorbidity. However, the opinion on the interaction of these two pathologies remains ambiguous. On the one hand, a plethora of studies demonstrate the impact of MAFLD on the progression of COVID-19. MAFLD worsens the COVID-19 course by disrupting innate immune response, altering the gut microbiota, changing the metabolic profile, and decreasing liver function. COVID-19 aggravates liver damage through direct virus-damaging effects, blood rheology disruption, tissue hypoxia, and drug-induced injury during viral infection treatment (Figure 2). Limitations in studies, the lack of genetic evidence, and studies showing the absence of an impact of MAFLD on the progression of COVID-19 prevent definitive conclusions. Additionally, the divergence in definitions between MAFLD and NAFLD reduces the precision of the ultimate estimation of their effect on the progression of COVID-19. We should take these limitations into account when planning further research, as this topic will continue to remain relevant in the future.

## Figures and Tables

**Figure 1 viruses-15-01072-f001:**
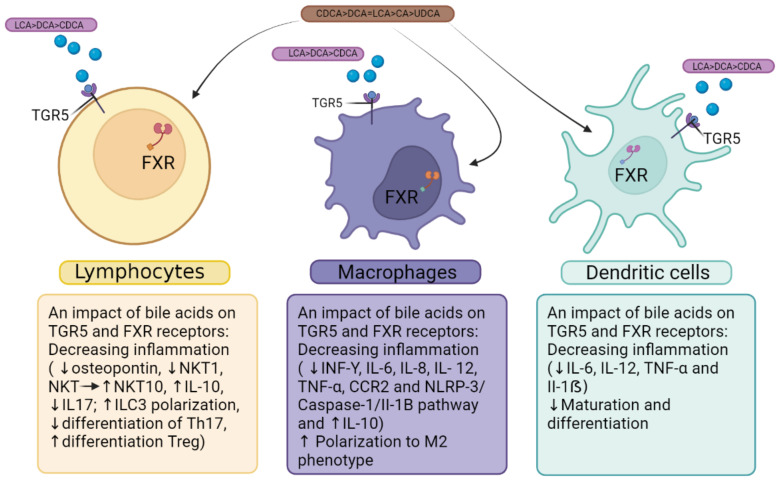
Functional role of Takeda G-protein-coupled receptor 5 (TGR5) and Farnesoid-X-receptor (FXR) in cells of immunity. The figure was created using BioRender.

**Figure 2 viruses-15-01072-f002:**
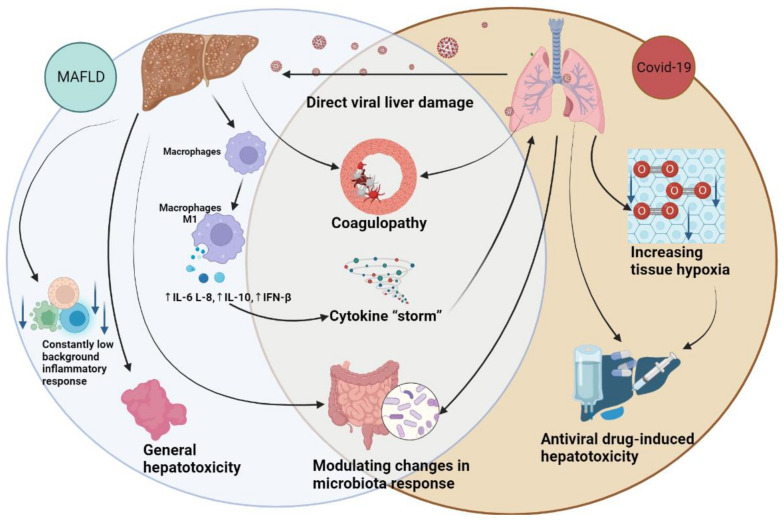
Pathogenetic ways of interaction between COVID-19 and MAFLD. The figure was created using BioRender.

**Table 4 viruses-15-01072-t004:** Shared alterations in the microbiota, metabolites, and inflammatory factors in COVID-19 and MAFLD.

MAFLD				COVID-19		
Gut Metabolites and Inflammatory Factors	Gut Microbiome Changes		Overlap	Gut Microbiome Changes		Gut Metabolites and Inflammatory Factors
↑ Endotoxins [157,159]: activating the TLR4; ↑ TNF-α, IL-1β, IL-6 and IL-12; hepatocyte injury; oxidative stress; hepatocyte apoptosis. ↑Lipopolysaccharides [160]: Activating of TLR4; ↑ IL-6,IL-1β, serum LBP TNF-α, chemokines; ↓↑ Bile acid metabolites [151,161,162,163]: ↑ IL-6,IL-8, IL-12,IL-1β, TNF-α, IL-1β, IFN-y; ↓ IL-10. ↑ Bacterial DNAs [164]: activating the TLR9; activating of NF-κB/MAPK; macrophages, NK cells, B cells, dendritic cells; ↑ IL-12 and TNF-α. ↑ Peptidoglycan [165]: activating of NF-κB/MAPK, NOD1, NOD2; ↑ pro-inflammatory cytokines.↓ Indole-3-acetic acid (IAA) [166]: ↑ TNF-α, MCP-1 тa IL-1β.	*↓ Alistipes* *↓ Anaerosporobacter* *↓ Coprobacter* *↓ Haemophilus* *↓ Moryella* *↓ Oscillobacter* *↓ Pseudobutyrivibrio* *↓ Subdoligramulum* *↓ Methanobrevibacter* *↓ Oscillospira* *↓ Phascolarctobacterium* [158] *↓ Rhuminococcaceae* *↓ Rikenellaceae* *↓ Prevotella* *↓ Prevotellaceae* *↓ Clostridiaceae* *↓ Clostridium* [167]	*↑ Acidaminococcus* *↑ Akkermansia* *↑ Allisonella* *↑ Anaerococcus* *↑ Bradyrhizobium* *↑ Dorea* *↑ Eggerthella* *↑ Escherichia* *↑ Flavonifractor incertae sedis* *↑ Parabacteroides* *↑ Peptoniphilus* *↑ Porphyromonas* *↑ Robinsonella* *↑ Ruminococcus* *↑ Shigella* [158] *↑ Proteobacteria* *↑ Enterobacteria* [168] *↑ Subdoligranulum* *↑ Blautia sp* *↑ Firmicutes* *↑ Roseburia* *↑ Oscillibacter* [169] *↑ Fusobacteria* [167]	*↑ Bacteroides, * *↓ Bifidobacterium* *↓ Eubacterium, * *↓ Faecalibacterium* [170]-COVID-19 [158]-MAFLD *↓ Coprococcus* [154]-COVID-19 [158]-MAFLD *↑ Streptococcus* [155]-COVID-19 [169]-MAFLD *↑ Enterobacteriaceae* [170]-COVID-19 [167]-MAFLD *↓ Lactobacillus* [171]-COVID-19 [158]-MAFLD	*↓ Roseburia,* *↓ Lachnospiraceae * *↓ Bacteroidetes* *↓ Blautia wexlerae,* [170] *↓ Dorea* *↓ Ruminococus* [154] *↓ Ruminococcus bromii* [171]	*↑ Enterobacteriaceae* *↑ Enterococcus* *↑ Actinomyces* *↑ Clostridium* [170] *↑ Rothia* *↑ Veillonella* [155] *↑ Blautia spp* *↑ Campylobacter* *↑ Corynebacterium* *↑ Enterococcaceae* *↑ Pseudomonas* *↑ Staphylococcus* [171] *↑ Klebsiella* [172]	↓ SCFA [173,174]: ↓ effector T cells; ↓ IL-17, IFN-γ, and/or IL-10. ↑ Lipopolysaccharides [150,160]: Activating of TLR4; ↑ IL-6,IL-1β, serum LBP TNF-α, chemokines; ↓↑ Bile acid metabolites [150,151,153,161,162,163] inhibit NF-Κb ↑ IL-6,IL-8, IL-12,IL-1β, TNF-α, IL-1β, IFN-y; ↓ IL-10; progression of respiratory failure [153]

LBP—lipopolysaccharide-binding protein; NF-κB—nuclear factor-kappa B; MAPK—mitogen-activated protein kinase; NOD1—Nucleotide Binding Oligomerization Domain Containing 1; NOD2—Nucleotide Binding Oligomerization Domain Containing 2; SCFA—short-chain fatty acids.

## Data Availability

Data is contained within the article.

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
