# Peer review of "The Intersection of COVID-19 and Metabolic-Associated Fatty Liver Disease: An Overview of the Current Evidence"

_viruses, 2023, doi:10.3390/v15051072_

Round 1

Reviewer 1 Report

Comments to Authors 

            This study showed that: a) changes in both innate and adaptive immune responses among metabolic dysfunction-associated fatty liver disease (MAFLD) patients may play a role in determining the severity of COVID-19; b) the remarkable similarities observed in the cytokine pathways implicated in both diseases imply the existence of shared mechanisms governing the chronic inflammatory responses characterizing these conditions.

          Authors are kindly requested to emphasize the current concepts about these issues in the context of recent knowledge and the available literature. This articles should be quoted in the References list.

References

1.      Gut Microbiota, NAFLD and COVID-19: A Possible Interaction. Obesities 2022, 2, 215–221. https://doi.org/10.3390/ obesities2020017.

2.      The Impact of COVID-19 Pandemic Lockdown on the Relationship between Pediatric MAFLD and Renal Function. J Clin Med. 2023; 12 (5): 2037. Published 2023 Mar 4. doi:10.3390/jcm12052037.

3.      COVID-19 and MAFLD/NAFLD: An updated review. Front Med (Lausanne). 2023; 10:1126491. Published 2023 Mar 24. doi:10.3389/fmed.2023.1126491.

 Minor editing of English language required

Author Response

We thank the Reviewer for the constructive comments. The most current concepts were supported. The relevant articles were cited.

Reviewer 2 Report

 The review article “The intersection of COVID-19 and MAFLD: An overview of the  current evidence” by  Mykhailo Buchynskyi et al. summarizes data about a possible link between MAFLD and COVID-19.

The article has to be rewritten. Whereas several descriptions are repeated (e.g. inflammation and insulin resistance) others are just mentioned e.g. IL-17-expressing helper T cells. The article has to be carefully corrected. Most studies refer to NAFLD and the use of NAFLD and MAFLD is confusing. This has to be better defined.

Abbreviations should not be used in the title.

NAFLD is not an appropriate keyword.

Identification of modifiable risk factors is essential to developing targeted prevention strategies for this condition.” Next sentence refers to age and male sex and this are not modifiable risk factors. Please rewrite.

“The prevalence of metabolic and vascular disorders …” Please also provide information about the abundance of these cases in the whole population.

Please explain abbreviations such as PAI-1

“The activation of proinflammatory cytokines, including TNFα, IL-6, and IL-1β, is increased in this cohort of patients. In patients with metabolic syndrome, these  cytokines are activated in adipose tissue” what does this mean – cytokines are activated?

The definition of MAFLD is not correct. “An intriguing aspect of this new definition, that exclusion of other significant alcohol intake or other chronic liver disease is not perquisite for the diagnosis of MAFLD anymore” https://doi.org/10.1111/liv.14478

cytokines and adipokines by hepatic macrophages” hepatic macrophages do not produce adipokines per definition

presence of liver  obesity“ ?

„nonalcoholic steatohepatitis (NAS)“ usually abbreviated as NASH as was done at page 14 “MAFLD to non-alcoholic steatohepatitis (NASH)”

Table 1; “Patients were only of Asian ethnicity“ this is a limitation of several of the studies listed.

“Nevertheless, MAFLD is a multi-system disorder and, unlike NAFLD, includes other metabolic disorders such as type 2 diabetes mellitus (T2DM), overweight/obesity, or other metabolic disturbances not related to alcohol consumption or other accompanying liver diseases [25].” This is completely wrong.

Table 3 needs explanations. What do the % mean? Sample size 722-757? How many of the controls have elevated ALT?

Page 13 “aspartate aminotransferase (AST) and alanine aminotransferase (ALT)” has to be defined when used the first time”

“systematic reviews demonstrated that elevated levels of cholestatic liver enzymes, including alkaline phosphatase (ALP) and gamma-glutamyltransferase (GGT), were present in 6.1% and 21.1% of COVID-19 patients, respectively” this has to be compared to the general population.

Mortality in transplant patients may be caused by intake of immunosuppressive drugs.

“moderate loss of its function can reduce the therapeutic efficacy of antiviral drugs that are metabolized in it.” This needs explanation, do the authors mean activation by the live or excretion?

“enters the cell by endocytosis, and the viral genome is released from the  endosome.” The virus can also enter the cell by fusion.

“However, the results of the study by Cai et al., 2020 [109] showed that in hypertensive patients receiving ACE inhibitors/angiotensin receptor blockers (ARBs), there was no increase in the frequency of COVID-19 progression to a severe form compared to patients  taking other antihypertensive drugs.” The authors should discuss this.

“Peptidoglycan, short-chain fatty acids (SCFA), Bile acid metabolites, Endotoxins, and several others,” lower case letter for e.g. endotoxins

Table 4 has to be improved for clarity.

“Intestinal dysbiosis can elicit inflammation in both the intestine and liver, which can  be attributed to the translocation of endotoxins and bacteria, resulting in rise in intestinal permeability.” May be higher intestinal permeability comes first.

“Thus, it can be assumed that the intestinal microbiota plays an important role in the 405 progression of MAFLD as well as the severity and mortality of COVID-19” was this confirmed. Altered microbiome may be the consequence of COVID-19. Was its role in disease progression confirmed?

Are bile acid levels changed in COVID-19? Proinflammatory effects of bile acids are not described.

Please carefully correct the article.

Author Response

We thank the Reviewer for the comments.

Comments from the editors and reviewers:

Reviewer 2:

Q1. The article has to be rewritten. Whereas several descriptions are repeated (e.g. inflammation and insulin resistance) others are just mentioned e.g. IL-17-expressing helper T cells. The article has to be carefully corrected. Most studies refer to NAFLD and the use of NAFLD and MAFLD is confusing. This has to be better defined.

Answer: The majority of studies utilized the definition of NAFLD rather than MAFLD. NAFLD is a more traditional term used to describe fatty liver disease caused by metabolic disorders, whereas MAFLD is a newer term that considers a broader range of metabolic and non-metabolic factors in the development and progression of fatty liver disease. We endeavored to reconcile these discrepancies in definitions while maintaining fidelity to the descriptions outlined in the relevant studies.

Q2. Abbreviations should not be used in the title.

Answer:. We have changed the title.

Q3. NAFLD is not an appropriate keyword.

Answer: We have removed NAFLD from keywords and added MALFD.

Q4. “ Identification of modifiable risk factors is essential to developing targeted prevention strategies for this condition.” Next sentence refers to age and male sex and this are not modifiable risk factors. Please rewrite.

Answer: We have changed this sentence.

Q5. “The prevalence of metabolic and vascular disorders …” Please also provide information about the abundance of these cases in the whole population.

Answer: We have added the information about the abundance of the prevalence of metabolic and vascular disorders in the whole population.

Q6. Please explain abbreviations such as PAI-1

Answer: We have added the abbreviation for PAI-1

Q7. “The activation of proinflammatory cytokines, including TNFα, IL-6, and IL-1β, is increased in this cohort of patients. In patients with metabolic syndrome, these  cytokines are activated in adipose tissue” what does this mean – cytokines are activated?

Answer: In obesity, the adipose tissue is a site of immune cell accumulation. This contributes to the increase in the synthesis of bioactive molecules such as TNFα, IL-6,  IL-1β, etc. We changed this fragment for better understanding.

Q8. The definition of MAFLD is not correct. “An intriguing aspect of this new definition, that exclusion of other significant alcohol intake or other chronic liver disease is not perquisite for the diagnosis of MAFLD anymore” https://doi.org/10.1111/liv.14478

 Answer: We changed the definition and added the necessary details.

Q9. “cytokines and adipokines by hepatic macrophages” hepatic macrophages do not produce adipokines per definition

Answer: According to research by Kazankov K, et al 2018, macrophages are also capable of producing adipokines (doi:10.1038/s41575-018-0082-x). The relevant link has been added. We changed this fragment.

Q10. “presence of liver obesity“ ?

Answer: We have corrected this part.

Q11. „nonalcoholic steatohepatitis (NAS)“ usually abbreviated as NASH as was done at page 14 “MAFLD to non-alcoholic steatohepatitis (NASH)”

Answer: We have corrected this abbreviation.

Q12. Table 1; “Patients were only of Asian ethnicity“ this is a limitation of several of the studies listed.

Answer: We have added needed limitations to such studies.

Q13. “Nevertheless, MAFLD is a multi-system disorder and, unlike NAFLD, includes other metabolic disorders such as type 2 diabetes mellitus (T2DM), overweight/obesity, or other metabolic disturbances not related to alcohol consumption or other accompanying liver diseases [25].” This is completely wrong.

 Answer: We removed this sentence.

Q14.Table 3 needs explanations. What do the % mean? Sample size 722-757? How many of the controls have elevated ALT?

Answer: In some studies, not all patients had all necessary blood parameters determined. Therefore, the control group is different in terms of the number of patients. The slash shows the number of patients with an increase in markers of liver damage, as opposed to others whose liver enzymes were normal. We have added more details to explain it.

Q15. Page 13 “aspartate aminotransferase (AST) and alanine aminotransferase (ALT)” has to be defined when used the first time”

Answer: We have put the definitions in the right order.

Q16. “systematic reviews demonstrated that elevated levels of cholestatic liver enzymes, including alkaline phosphatase (ALP) and gamma-glutamyltransferase (GGT), were present in 6.1% and 21.1% of COVID-19 patients, respectively” this has to be compared to the general population.

Answer: These data are absent in the analyzed articles

Q17. Mortality in transplant patients may be caused by intake of immunosuppressive drugs.

Answer: We changed this fragment and added the information.

Q18. “moderate loss of its function can reduce the therapeutic efficacy of antiviral drugs that are metabolized in it.” This needs explanation, do the authors mean activation by the live or excretion?

Answer: We have added the explanation.

Q19. “enters the cell by endocytosis, and the viral genome is released from the  endosome.” The virus can also enter the cell by fusion.

Answer: We have added the explanation.

Q20. “However, the results of the study by Cai et al., 2020 [109] showed that in hypertensive patients receiving ACE inhibitors/angiotensin receptor blockers (ARBs), there was no increase in the frequency of COVID-19 progression to a severe form compared to patients  taking other antihypertensive drugs.” The authors should discuss this.

Answer: We have added the information, that displays our opinion.

Q21. “Peptidoglycan, short-chain fatty acids (SCFA), Bile acid metabolites, Endotoxins, and several others,” lower case letter for e.g. endotoxins

Answer: We have corrected the text according to your suggestion.

Q22. Table 4 has to be improved for clarity.

Answer:  Table 4 was modified according to your suggestions.

Q23. “Intestinal dysbiosis can elicit inflammation in both the intestine and liver, which can  be attributed to the translocation of endotoxins and bacteria, resulting in rise in intestinal permeability.” May be higher intestinal permeability comes first.

Answer: We have rephrased this sentence.

Q24. “Thus, it can be assumed that the intestinal microbiota plays an important role in the progression of MAFLD as well as the severity and mortality of COVID-19” was this confirmed. Altered microbiome may be the consequence of COVID-19. Was its role in disease progression confirmed?

Answer: Based on previous researches included in our review, we can assume such an impact. We need more research in this area. We have added the information that displays our opinion.

Q25. Are bile acid levels changed in COVID-19? Proinflammatory effects of bile acids are not described.

Answer: Yes, secondary bile acids are decreased in COVID-19 patients (10.1038/s41467-022-34260-2). The pro-inflammatory effect of bile acids is mainly related to a decrease in their level. In this section, we described the effect that bile acids exert directly on the immune response, which is altered in COVID-19 and MAFLD. We have added the information.

Reviewer 3 Report

Buchynskyi et al presented the relationship of COVID-19 and MAFLD. But they changed the focus abruptly to NAFLD and liver injury. I think writings should be organized and corrected before publication of this review in title, abstract and introduction.

1.       As it is generally considered the appearance of COVID-19 in Dec; 2019, it should be maintained. If authors wish to keep March 20, they should mention in Eukraine.

2.       Line 122, authors should explain what is “in this  cohort of patients” in the text or omit it.

3.       Line 203, it would be  diagnosed with MAFLD.

4.       Line 216-217, Already abbreviated as NAFLD, no need to write full condition here.

5.       It seems authors lost the path. Suddenly they shifted their attention to NFALD instead of MFALD. If they want to discuss NFALD also, they should discuss the relationship between two in introduction first.

6.       The purpose of this review turns into MFAL, NFALD and liver injury. Authors presented MFALD, NFALD and liver injury without mentioning at all in abstract and introduction. So the significance is lost.

7.       Moderate English grammar and editing are needed.

Moderate editing required.

Author Response

We thank the Reviewer for the constructive comments.

Q1. As it is generally considered the appearance of COVID-19 in Dec; 2019, it should be maintained. If authors wish to keep March 20, they should mention in Eukraine.

Answer: We have corrected this part.

Q2. Line 122, authors should explain what is “in this cohort of patients” in the text or omit it.

Answer: We have rephrased this sentence.

Q3. Line 203, it would be diagnosed with MAFLD.

Answer: We have changed the sentence.

Q4. Line 216-217, Already abbreviated as NAFLD, no need to write full condition here.

Answer: We have corrected this abbreviation.

Q5.  It seems authors lost the path. Suddenly they shifted their attention to NFALD instead of MFALD. If they want to discuss NFALD also, they should discuss the relationship between two in introduction first.

Answer: MAFLD is a newer name for NAFLD. The terms MAFLD and NAFLD describe the same condition. International experts suggest that the term MAFLD more accurately represents the underlying pathogenesis of the disease compared to the previously used term, NAFLD (DOI: https://doi.org/10.1016/S2468-1253(22)00062-0, doi:10.1111/LIV.14478). Despite the slight differences between these definitions mentioned in the introduction, the two terms are essentially identical. The majority of studies have utilized the NAFLD definition. We tried to reconcile the discrepancies between the definitions while remaining faithful to the descriptions provided in relevant studies. All references were included in the article.

Q6.  The purpose of this review turns into MFAL, NFALD and liver injury. Authors presented MFALD, NFALD and liver injury without mentioning at all in abstract and introduction. So the significance is lost.

Answer: In the introduction, we discussed the terminology of NAFLD and MAFLD. We have added new information.

Q7.  Moderate English grammar and editing are needed.

Answer: The grammar has been corrected.

Round 2

Reviewer 2 Report

Figure 1 has to be corrected, decreasing inflammatory ?

Bracket is missing in the right box

Table 2 please correct Refferences

English is o.k.

Author Response

We thank the Reviewer for the constructive comments.

We have corrected Figure 1 and the References in Table 2.